# CARs and Drugs: Pharmacological Ways of Boosting CAR-T-Cell Therapy

**DOI:** 10.3390/ijms24032342

**Published:** 2023-01-25

**Authors:** Dennis Christoph Harrer, Jan Dörrie, Niels Schaft

**Affiliations:** 1Department of Hematology and Internal Oncology, University Hospital Regensburg, 93053 Regensburg, Germany; 2Department of Dermatology, Friedrich-Alexander-Universität Erlangen-Nürnberg, Universitätsklinikum Erlangen, Hartmannstraße 14, 91052 Erlangen, Germany; 3Comprehensive Cancer Center Erlangen European Metropolitan Area of Nuremberg (CCC ER-EMN), Östliche Stadtmauerstraße 30, 91054 Erlangen, Germany; 4Deutsches Zentrum Immuntherapie (DZI), Ulmenweg 18, 91054 Erlangen, Germany

**Keywords:** drugs, immunotherapy, pharmaceuticals, microenvironment, combination

## Abstract

The development of chimeric antigen receptor T cells (CAR-T cells) has marked a new era in cancer immunotherapy. Based on a multitude of durable complete remissions in patients with hematological malignancies, FDA and EMA approval was issued to several CAR products targeting lymphoid leukemias and lymphomas. Nevertheless, about 50% of patients treated with these approved CAR products experience relapse or refractory disease necessitating salvage strategies. Moreover, in the vast majority of patients suffering from solid tumors, CAR-T-cell infusions could not induce durable complete remissions so far. Crucial obstacles to CAR-T-cell therapy resulting in a priori CAR-T-cell refractory disease or relapse after initially successful CAR-T-cell therapy encompass antigen shutdown and CAR-T-cell dysfunctionality. Antigen shutdown predominately rationalizes disease relapse in hematological malignancies, and CAR-T-cell dysfunctionality is characterized by insufficient CAR-T-cell proliferation and cytotoxicity frequently observed in patients with solid tumors. Thus, strategies to surmount those obstacles are being developed with high urgency. In this review, we want to highlight different approaches to combine CAR-T cells with drugs, such as small molecules and antibodies, to pharmacologically boost CAR-T-cell therapy. In particular, we discuss how certain drugs may help to counteract antigen shutdown and CAR-T-cell dysfunctionality in both hematological malignancies and solid tumors.

## 1. Introduction

Over the last decade, cellular immunotherapy has emerged as a new powerful therapeutic pillar in the battle against cancer. Striking success was attained in patients suffering from advanced hematological malignancies by the use of T cells genetically modified to express a chimeric antigen receptor (CAR) [1,2,3,4,5]. In light of high response rates with a sizable proportion of long-lasting complete remissions in ill-fated patients with refractory lymphoma/leukemia, FDA and EMA approval was issued to several CAR-T-cell products for treating acute lymphoblastic leukemia (ALL), diffuse large B cell lymphoma (DLBCL), follicular lymphoma (FL), mantle cell lymphoma (ML), and multiple myeloma (MM) [6]. Moreover, a myriad of clinical trials are currently running to evaluate the efficacy of CAR-T cells against multiple other tumor entities beyond lymphoma/leukemia [7,8]. 

Nevertheless, approximately 50% of lymphoma patients treated with commercially available CAR-T-cell products either are a priori refractory to CAR-T-cell therapy or show relapsing disease after initial responses to CAR-T cells [9]. In addition, the lion’s share of patients suffering from solid tumors, such as prostate carcinoma or pancreatic carcinoma, did not exhibit durable remissions in response to CAR-T-cell therapy so far. Important hurdles equally obstructing CAR-T-cell efficacy in hematological malignancies and solid tumors include antigen shutdown and CAR-T-cell dysfunctionality. Disease relapse after CAR-T-cell treatment in hematological malignancies is primarily predicated on a reduction in antigen density originating from various mechanisms of antigen shutdown [10]. In patients with solid tumors, primary CAR-T-cell failure is frequently observed with lacking initial responses and incessant disease progression after CAR-T-cell infusion [11]. A leading cause of the inefficacy of CAR-T cells in solid tumors could be traced back to CAR-T-cell dysfunctionality emerging rapidly after CAR-T-cell infusion [11]. Characteristic manifestations of CAR-T-cell dysfunctionality encompass insufficient CAR-T-cell proliferation coupled with impaired cytokine secretion and cytotoxicity [12]. Moreover, solid tumors are frequently embedded within an immunosuppressive tumor microenvironment consisting of regulatory T cells, myeloid-derived suppressor cells, macrophages, endothelial cells, and fibroblasts [13,14]. This shielding tumor microenvironment inhibits T-cell migration, invasion, and antitumor activity, thereby posing a crucial obstacle to CAR-T-cell efficacy in solid tumors [13,14]. 

In order to boost CAR-T-cell performance, combination strategies with drugs, such as antibodies or small molecules, have gained attention. Synthetic biology has brought forward powerful next-generation CAR-T-cell products that are considerably less prone to CAR-T-cell dysfunctionality [12] or can target tumor cells with very low numbers of antigen on the surface [15]. Nevertheless, approaches to simultaneously tackle CAR-T-cell dysfunction, antigen shutdown, and the immunosuppressive tumor microenvironment are scant. Moreover, due to constraints of current genetic engineering approaches, such as viral vector size, the synthetic complexity that can be added to CAR-T cells is limited. Hence, pharmacological ways to boost CAR-T-cell therapy using drugs, which could be easily coadministered with CAR-T cells via intravenous infusion or oral intake, could obviate challenges of CAR-T-cell products without requiring complex genetic engineering. In theory, a plethora of different combinations of CAR-T cells and various drugs are conceivable depending on the tumor entity, the CAR-T-cell product used, and the characteristics of the individual patient. 

In this review, we want to shine a light on different approaches to combine CAR-T cells with drugs to pharmacologically boost CAR-T-cell therapy. We discuss how certain drugs may aid to obviate prominent obstacles to CAR-T-cell efficacy in both hematological malignancies and solid tumors. Drug-based strategies to mitigate toxicity of CAR-T cells are not part of this review and have been extensively discussed elsewhere. 

## 2. Pharmacological Ways of Boosting CAR-T-Cell Therapy

### 2.1. Hematological Malignancies

T cells engineered to target CD19 on lymphoma and leukemia cells were the first successful CAR-T-cell product to be clinically used more than a decade ago [16]. In recent years, CAR-T-cell products were approved for the treatment of multiple myeloma, and clinical evaluation of CAR-T-cell therapy in acute myeloid leukemia (AML) and T-cell lymphoma is currently gaining momentum [6]. Common issues observed in CAR-T-cell therapy of hematological malignancies are antigen shutdown and CAR-T-cell dysfunctionality, frequently attributed to a progressive loss of canonical T-cell functions termed T-cell exhaustion [12,17]. Thus, pharmaceuticals could be beneficial to combat antigen shutdown via upregulation of CAR-T-cell target antigens on tumor cells or via combinatorial CAR-independent targeting of tumor cells. Furthermore, drugs could be useful to improve overall CAR-T-cell functionality by interfering with pathways entangled in T-cell exhaustion and negative feedback regulation.

#### 2.1.1. Upregulation of Target Antigens

In hematological malignancies, several drugs have been identified to enhance the reactivity of CAR-T cells by the upregulation of CAR-target antigens on B cell lymphoma/leukemia cells. Seeking to boost the notoriously modest in vivo activity of CAR-T cells expressing a CD27-based CAR against acute myeloid leukemia cells expressing the CAR target antigen CD70 at mediocre levels, Leick et al. adopted a pharmacological approach predicated on the demethylating agent azacitidine to enhance the density of CD70 on AML cells [18] (Figure 1). Malignant myeloid cells treated with azacitidine showed increased CD70 expression and were more prone to CAR-mediated elimination [18]. Analogously, azacitidine, which is usually employed in the treatment of myelodysplastic syndrome or acute myeloid leukemia, could upregulate CD123 expression on leukemic cells, facilitating recognition and destruction by CD123-specific CAR-T cells [19]. Furthermore, decitabine, which is a methyltransferase inhibitor similar to azacitidine, was shown to increase CD19 expression on B-cell lymphoma cell lines without impairing CAR-T-cell functionality [20] (Figure 1). In light of those results, two patients with relapsing/refractory B-cell lymphoma received pretreatment with decitabine before infusion of CD19-specific CAR-T cells, leading to complete remission in both patients [21]. Existing data support the further exploration of combining CAR-T cells with demethylating agents, especially in cases of antigen-negative or antigen-dim lymphoma relapse.

The heterogenous expression of B-cell maturation antigen (BCMA), which is the primary CAR target in multiple myeloma, and BCMA shutdown under CAR-T-cell pressure pose crucial issues in the CAR-T-cell therapy of multiple myeloma [22,23]. Using the epigenetic modulator all-trans retinoic acid (ATRA) (Figure 1), BCMA expression could be augmented in multiple myeloma cell lines and primary multiple myeloma cells [24]. Additionally, the pharmacological upregulation of BCMA in myeloma cells eventuated in enhanced reactivity of BCMA-CAR-T cells in vitro and in vivo [24]. A combined treatment of myeloma cells with ATRA and the γ-secretase inhibitor crenigacestat (Figure 1) could further increase BCMA expression, inducing even stronger antigen-specific targeting by BCMA-specific CAR-T cells [24]. The rationale for the use of γ-secretase inhibitors is derived from the active cleavage of BCMA from the surface of myeloma cells by the ubiquitous γ-secretase (GS) complex [24]. Importantly, soluble BCMA is capable of CAR-T-cell activation. Exposure of multiple myeloma-bearing immunodeficient mice to γ-secretase inhibitors augmented BCMA expression on malignant cells, reduced the presence of soluble BCMA, and promoted the antitumor activity of BCMA-specific CAR-T cells [25]. 

Another approach to enhance antigen expression on myeloid leukemia cells was suggested by Jetani et al., laying the framework for combining CAR-T cells with small molecule inhibition [26]. Upon treatment with the kinase inhibitor crenolanib (Figure 1), which blocks the activity of the oncogenic transmembrane protein FMS-like tyrosine kinase 3 (FLT3) on leukemia cells, augmented surface expression of FLT3 could be observed specifically on FLT3-ITD+ AML cells [26]. Correspondingly, AML cells with crenolanib-upregulated FLT3 expression were more effectively recognized by FLT3-specific CAR-T cells [26]. 

Finally, the histone deacetylase inhibitor (HDACi) chidamide (Figure 1), which is capable of broadly modulating gene expression in B-cell lymphoma cell lines, was found to significantly upregulate CD22 expression in lymphoma cell lines in vitro and in vivo [27]. Moreover, the chidamide-mediated upregulation of CD22 on lymphoma cells intensified the antitumor response from CD22-specific CAR-T cells [27]. Similarly, the protein kinase C inhibitor bryostatin I could invigorate CD22 expression on lymphoid malignant cells, bolstering recognition by CD22-specific CAR-T cells [28]. 

#### 2.1.2. Combinatorial Targeting

Apart from the upregulation of target antigens, drugs can further boost CAR-T-cell therapy by directly engaging CAR-independent target structures in tumor cells, such as regulators of apoptosis. Overexpression of the antiapoptotic molecule B-cell lymphoma 2 (BCL-2) in B-cell lymphoma/leukemia cells is commonly associated with decreased patient survival [29,30]. Seeking to simultaneously target lymphoma cells with CD19-specific CAR-T cells and the FDA-approved BCL2 inhibitor venetoclax (Figure 2), investigators led by Marco Ruella designed CAR-T cells with an intrinsic resistance to venetoclax-mediated apoptosis induction by overexpressing a mutated BCL2 version in CD19-specific CAR-T cells (BCL-2(F104L)-CART19) [31]. The combination of venetoclax and CAR-T cells displayed synergy in several murine lymphoma models without impairing the functionality or persistence of CAR-T cells [31]. Without engineered venetoclax resistance, the simultaneous or subsequent administration of BH3 mimetics, such as venetoclax or the Mcl-1 inhibitor S63845 (Figure 2), together with CAR-T cells significantly reduced the survival of the CAR-T-cell product. Hence, Yang et al. identified the presensitization of tumor cells with BH3 mimetics followed by the infusion of CAR-T cells as the most efficacious combinatorial approach to enhance the overall elimination of tumor cells [32]. Nevertheless, conflicting data exist advocating the concurrent administration of CAR-T cells and venetoclax due to synergy in killing rituximab-resistant B-cell lymphoma cells, decrease of regulatory T cells, and promotion of T-cell stemness [33]. Equal to venetoclax, the BCL-2 inhibitor ABT-737 (Figure 2) was found to potentiate CAR-T-cell cytotoxicity against selected B-cell lymphoma cell lines resistant to apoptosis, underscoring the advantage of combinatorial targeting of CAR-T cells and small molecules [34]. 

Before the advent of CAR-T-cell therapy, antigen-specific targeting of B-cell lymphoma/leukemia cells was based on antibodies, such as the CD20-directed antibodies rituximab and obinutuzumab [35]. While CAR-T cells are directly capable of killing tumor cells, antibodies mark tumor cells for secondary destruction by natural killer cells, macrophages, or complement-mediated cell death (Figure 3). Given the laborious and expensive production of CAR-T cells, the concomitant administration of readily available off-the-shelf antibodies could facilitate combinatorial targeting of different surface molecules on tumor cells. Currently, an ongoing clinical trial is evaluating the concurrent infusion of CD19-specific CAR-T cells together with rituximab in patients with refractory B-cell lymphoma (NCT04002401). 

Another approach to boost the activity of CD19-specific CAR-T cells against B-cell lymphoma cells with enhanced resistance to CAR-T-cell cytotoxicity relies on the selective cyclooxygenase (COX)-2 inhibitor celecoxib (Figure 2), which could substantially restore the cytotoxicity of CAR-T cells to lymphoma cells by partially reversing the resistance to TRAIL-mediated apoptosis [36]. In contrast, data demonstrating not only apoptosis induction in lymphoma cells by etoricoxib, but also enhanced exhaustion and attrition of CAR-T cells upon coincubation with etoricoxib, pose a caveat to combinatorial targeting with COX inhibitors [37]. 

#### 2.1.3. Improving CAR-T-Cell Functionality 

In addition to the upregulation of target antigens or combinatorial targeting, drugs can be used to directly augment the functionality of CAR-T cells. Ibrutinib, an inhibitor of Bruton’s tyrosine kinase (BTK) and IL-2-inducible T-cell kinase (ITK) (Figure 4), was one of the first drugs ever to be used alongside CAR-T cells, based on observations that ibrutinib could extend the in vivo persistence of T cells and reduce the immunosuppressive profile of lymphoma cells [38,39]. Correspondingly, the concurrent administration of ibrutinib and CD19-specific CAR-T cells could improve CAR-T-cell engraftment, resulting in enhanced tumor clearance and survival in murine models of resistant ALL and chronic lymphocytic leukemia (CLL) [39]. Thereafter, a pilot clinical trial on the simultaneous application of ibrutinib and CD19-specific CAR-T cells in heavily pretreated patients with refractory CLL was conducted [40]. The combination therapy was well tolerated and resulted in high rates of minimal residual disease (MRD)–negative complete remissions with superior 1-year progression-free survival as compared with CAR-T-cell treatment without ibrutinib [40]. In an ensuing phase II clinical trial, humanized CD19-specific CAR-T cells were coadministered together with ibrutinib to patients suffering from ibrutinib-refractory CLL [41]. This combination therapy could achieve long-lasting MRD-negative complete remissions with frequent but manageable manifestations of cytokine release syndrome [41]. Besides CLL, combining CAR-T cells with ibrutinib showed success in patients with mantle cell lymphoma and follicular lymphoma [42]. Remarkably, the novel selective BTK inhibitor acalabrutinib could bolster the functionality of CD19-specific CAR-T cells to a similar degree as ibrutinib, but only CAR-T cells treated with the dual BTK and ITK inhibitor ibrutinib evolved into memory-like T cells showing Th1 differentiation and preserved effector functions [43].

At the beginning of the adoptive T-cell therapy era, the coinjection of cytokines, such as IL-2, was a vital prerequisite to sustain the expansion and survival of transferred T cells, which were usually derived from in vitro amplified tumor-infiltrating lymphocytes (TILs) [44]. On the contrary, commercially available CAR-T-cell products usually do not require cytokine support after CAR-T-cell infusion, but benefit from elevated endogenous levels of the homeostatic cytokines IL-7 and IL-15 created by preceding lymphodepletion [45]. Owing to the short half-time of IL-15 necessitating frequent injections, the majority of approaches exploiting IL-15 signaling to boost CAR-T-cell persistence rely on engineering CAR-T cells with components of the IL-15/IL-15 receptor pathway [46,47,48] (Figure 5). 

In order to facilitate the clinical application of IL-15 injections, a polymer-conjugated version of IL-15 with a reduced drug clearance was investigated. In xenograft models of aggressive B-cell lymphoma, the coinjection of this stabilized version of IL-15 together with CD19-specific CAR-T cells augmented CAR-T-cell performance eventuating in the longer survival of tumor-bearing mice [49]. Mechanistically, IL-15 supplementation increased CAR-T-cell proliferation, cytokine production, and persistence by STAT5-dependent signaling [49]. Recently, data on a long-acting format of human recombinant IL-7 fused with hybrid Fc (rhIL-7-hyFc) were published (Figure 5). Similar to IL-15, injection of long-acting IL-7 together with CAR-T cells could extend the survival of tumor-bearing animals in both xenograft and immunocompetent mouse models [50]. Another elegant way of providing cytokines is by genetically manipulating the CAR-T cells to produce these cytokines themselves after antigen recognition, resulting in a locally very focused secretion of these cytokines, circumventing systemic side effects or the short half-life of these cytokines (Figure 5). Several clinical trials (e.g., NCT03542799, NCT03721068, NCT05103631, NCT04715191, NCT03198546, NCT02498912, and NCT03932565) used these kinds of CAR-T cells producing IL-12, IL-15, IL-21, or IL-7 on site. 

Immune checkpoint blockade aims at boosting tumor-specific T cells by breaking immunoevasion mediated by inhibitory checkpoint molecules, such as programmed cell death-1 (PD-1)/PD-ligand 1 (PD-L1). After promising preclinical results demonstrated that the blockade of PD-1 could increase the in vivo efficacy of CD19-specific CAR-T cells [51,52,53], seminal clinical experience on the combination of checkpoint blocking antibodies and CAR-T cells (Figure 5) was obtained in a B-cell lymphoma patient who evinced progressive lymphoma despite treatment with CD19-specific CAR-T cells [54]. In response to the infusion of pembrolizumab, a re-expansion of CD19-specific CAR-T cells was observed in peripheral blood coupled with systemic lymphoma regression [54]. Thereafter, several clinical trials were launched to systematically analyze the joint application of CAR-T cells and checkpoint blockade. First, 12 patients suffering from relapsed/refractory B-cell lymphoma after previous infusion of CD19-specific CAR-T cells received salvage therapy with the anti-PD1 antibody pembrolizumab at a median time of 3.3 months (range, 0.4–42.8 months) from CAR-T-cell infusion [55]. Clinical benefit was observed in 4 patients, all of whom displayed an increase in the percentage of circulating CAR-T cells coupled with enhanced CAR-T-cell functionality and less T-cell exhaustion [55]. No serious side effects were stirred by the use of pembrolizumab after the infusion of CD19-specific CAR-T cells [55]. In the currently ongoing phase I Alexander trial (NCT03287817), the safety and efficacy of novel CD19/CD22-specific CAR-T cells are analyzed in the context of short-term PD-1 blockade using pembrolizumab [56]. In an early interim analysis, approximately half of the treated patients with aggressive B-cell lymphoma entered into ongoing complete remission [56]. So far, no serious toxicities occurred. In a second trial, 11 patients with refractory/relapsed B-cell lymphoma were treated with CD19-specific CAR-T cells and the PD-1-blocking antibody nivolumab [57]. The addition of nivolumab to CAR-T-cell therapy did not induce serious toxicities and mediated a complete remission rate of 45.45% (5/11 patients) [57]. Additionally, the combination of nivolumab in a reduced dose (1.5 mg/kg) together with CD19-specific CAR-T cells showed efficacy as salvage therapy in a patient with refractory follicular lymphoma and high PD-1 expression on circulating T cells (80.90%) [58]. Furthermore, the PD-L1-blocking antibody atezolizumab displayed a manageable safety profile when administered together with CD19-specific CAR-T cells [59]. Preliminary data from this ZUMA-6 trial showed robust CAR-T-cell expansion and an objective response rate of 90% in pretreated patients with aggressive B-cell lymphoma [59]. Similar results were obtained in a small cohort of B-cell lymphoma patients cotreated with CD19-specific CAR-T cells and the PD-L1-blocking antibody durvalumab [60]. Apart from aggressive B-cell lymphoma, joint treatment with CAR-T cells and checkpoint blockade was also tested in ALL. Fourteen days after infusion of CD19-specific CAR-T cells, 11 juvenile ALL patients were boosted with pembrolizumab or nivolumab with optional repetition of PD-1 blocker infusion every 3 weeks [61]. Whereas early loss of CAR-T cells reflected by early B-cell recovery and bulky extramedullary disease emerged as predictive factors for response to PD-1 inhibition, relatively little efficacy was observed in patients refractory to initial CAR-T-cell treatment [61]. In multiple myeloma, bispecific CAR-T cells targeting BCMA and CS1 exhibited accelerated clearance of tumor cells in several mouse models when combined with a PD-1-blocking antibody [62]. In aggregate, PD-1/PD-L1 blockade appears to be a promising tool for counteracting acquired CAR-T-cell dysfunctionality in various hematological malignancies. The blockade of other checkpoint molecules, such as cytotoxic T lymphocyte–associated antigen-4 (CTLA-4), in the context of CAR-T-cell therapy is currently under clinical investigation (NCT00586391). Finally, the presence of T-cell immunoreceptor with Ig and ITIM domains (TIGIT) on T cells was found to be associated with CAR-T-cell dysfunction in patients diagnosed with B-cell lymphoma, and TIGIT blockade alone was sufficient to improve CAR-T-cell functionality in preclinical in vitro studies and mouse models [63]. 

In an effort to prolong the in vivo persistence of CAR-T cells by preventing differentiation into short-lived effector T cells, JQ1, an inhibitor of bromodomain and extraterminal motif (BET) proteins (Figure 4), was identified through comprehensive screening of epigenetic regulators to preserve stem-cell-like and central memory phenotypes in CD8-positive T cells [64]. In murine tumor models, CAR-T cells treated with JQ1 evinced enhanced persistence and antitumor efficacy [64]. Effector differentiation in CD8-positive T cells was reduced by JQ1-mediated inhibition of the BET protein BRD4, resulting in the reduced expression of the transcription factor BATF, which has been implicated in terminal effector cell differentiation and T-cell exhaustion [64]. 

Lenalidomide, an immunomodulatory drug with direct cytotoxic effects against myeloma cells, constitutes a major pillar of conventional multiple myeloma therapy. Combined with BCMA-specific CAR-T cells, lenalidomide promoted CAR-T-cell in vivo expansion (Figure 4), and significantly extended the survival of mice in a model of disseminated multiple myeloma [65]. Moreover, lenalidomide potentiated CAR-T-cell effector functions in a concentration-dependent manner and maintained long-term CAR-T-cell functionality during chronic antigen stimulation by delaying the development of functional T-cell exhaustion [65]. Similar results were reported regarding the combination of CS1-specific CAR-T cells and lenalidomide in mouse models of multiple myeloma, underscoring the potential of lenalidomide to serve as a universal booster for CAR-T-cell functionality in multiple myeloma [66]. The first successful clinical experience on the combination of BCMA-specific CAR-T cells and lenalidomide was reported in a patient with refractory IgD-λ multiple myeloma, who displayed disease progression despite multiple lines of therapy, including allogeneic stem cell transplantation and treatment with T cells expressing mouse-derived and human-derived BCMA-specific CARs [67]. After concurrent application of lenalidomide and humanized BCMA-CAR-T cells, the patient entered into a very good partial response, which could be maintained for at least 8 months [67]. No serious toxicities occurred. In an early-phase exploratory trial enrolling patients with high-risk newly diagnosed multiple myeloma, CD19-specific CAR-T cells and BCMA-specific CAR-T cells were sequentially infused after autologous stem cell transplantation, followed by lenalidomide maintenance therapy [68]. Seven out of 10 patients showed long-lasting MRD-negative complete remission ongoing for more than 2 years [68]. Another clinical trial evaluating the combination of lenalidomide and BCMA-specific CAR-T cells is currently running (NCT03070327). In sum, those data drawn from preclinical studies and a single case report provide the rationale for future clinical exploration of combining CD19-specific CAR-T cells with lenalidomide. Currently, the phase II multicenter ZUMA-14 trial is investigating the efficacy of combining CD19-specific CAR-T cells with lenalidomide (NCT04002401).

### 2.2. Solid Tumors

In contrast with hematological malignancies, where CAR-T-cell products have become a standard-of-care treatment, the clinical efficacy of CAR-T-cell therapy in patients with solid tumors lags behind. Despite numerous preclinical and clinical studies, CAR-T-cell products targeting solid tumors so far failed to obtain FDA approval. On the one hand, CAR-T-cell therapy in solid tumors faces similar obstacles as in hematological malignancies, such as antigen shutdown and CAR-T-cell dysfunctionality [14,69,70]. On the other hand, solid tumors pose unique challenges to CAR-T cells, such as the lack of suitable target antigens and the presence of a tumor-protective immunosuppressive microenvironment [14,69]. Hence, pharmaceuticals could aid CAR-T-cell therapy of solid tumors by upregulating CAR-T-cell target antigens to support antigen-specific targeting or by combinatorial CAR-independent targeting. Furthermore, drugs could be employed to boost overall CAR-T-cell functionality by preventing functional exhaustion. Finally, pharmacological targeting of the immunosuppressive microenvironment could both enhance the vulnerability of tumor cells to CAR-T-cell killing and prolong the survival of adoptively transferred T cells within the tumor complex. 

#### 2.2.1. Upregulation of Target Antigens

Unlike CD19 or BCMA in lymphoma/leukemia, which are uniformly expressed on tumor cells and are only coexpressed by dispensable cell types, such as normal B cells, the expression of potential CAR target antigens in solid tumors is more heterogeneous and frequently not restricted to tumor cells or dispensable cell types [14]. For instance, HER2 shows variable expression in carcinomas and glioblastoma, but is also found in lung epithelial cells, which caused lethal pulmonary toxicity in a patient receiving HER2-specific CAR-T cells for the treatment of colon cancer metastatic to the lungs and liver [71,72]. Of note is that the binding moiety of this CAR was based on the antibody trastuzumab, which is widely used in the treatment of HER2-positive breast cancer, and it did not cause these severe side effects. This indicates the potency but also the potential danger of CAR-T cells specific for certain antigens in solid tumors. Hence, the setup of a broad armamentarium of different target antigens is essential. In a recent preclinical study, the hypomethylating drug decitabine (Figure 1) could be exploited to upregulate the well-established melanoma antigen chondroitin sulfate proteoglycan 4 (CSPG4) in originally CSPG4-negative ovarian cancer cells [73]. Whereas in hematological malignancies and medulloblastoma, decitabine and azacitidine could enhance the expression of a priori expressed target antigens [70,74], the decitabine-mediated de novo upregulation of a CAR target antigen in carcinoma cells has never been reported before. After refining a protocol enabling the decitabine-mediated conversion of more than half of initially CSPG4-negative ovarian cancer cells to CSPG4-positive cells, CSPG4-directed cytokine secretion and cytotoxicity by CSPG4-specific CAR-T cells could be demonstrated [73]. Given the relative paucity of suitable CAR target antigens, decitabine-mediated upregulation of potential CAR target antigens should be assayed in different tumor entities. In parallel, thorough safety studies are required to detect the decitabine-mediated upregulation of CAR target antigens on healthy tissues to minimize the risk for on-target/off-tumor toxicity. 

#### 2.2.2. Combinatorial Targeting

Combinatorial targeting signifies a multi-hit attack on tumor cells through CAR-dependent signaling and a drug-based CAR-independent attack on other key structures of malignant cells, such as oncogenic protein kinases. Small molecules targeting receptor tyrosine kinases or mutated versions of cytoplasmic kinases have grown into major therapeutic pillars in various cancer entities (e.g., lung cancer, colon cancer, breast cancer, and malignant melanoma). However, data on a potential combinatorial targeting using the combination of kinase inhibitors and CAR-T cells are scant. In malignant melanoma with BRAF^V600E^ mutation, kinase inhibition employing BRAF and MEK inhibitors (BRAFi/MEKi) assumes an important role. In order to pave the way for combinatorial targeting of BRAFi/MEKi together with CAR-T cells, the BRAFi/MEKi combinations dabrafenib (Dabra) and trametinib (Tram) vs. vemurafenib (Vem) and cobimetinib (Cobi) were assayed for drug-induced alterations of CAR-T-cell activation and functionality [75] (Figure 3). Upon antigen-specific challenge in the presence of BRAFi/MEKi, the combination of Vem + Cobi significantly impaired CAR-T-cell activation and functionality to a higher degree than Dabra + Tram, which reduced cytokine secretion in CAR-T cells but had no tangible impact on CAR-T-cell cytotoxicity [75]. Thus, Dabra + Tram emerged as a favorable partner for combinatorial targeting with CAR-T cells in this proof-of-concept study [75]. Next, in vivo studies are required to further explore the therapeutic potential of Dabra + Tram combined with CAR-T cells in models of melanoma. Future research should aim at identifying further combination partners for CAR-T cells to establish a board arsenal for combinatorial targeting. 

#### 2.2.3. Improving CAR-T-Cell Functionality

In solid tumors, CAR-T-cell functionality is hampered by insufficient CAR-T-cell proliferation and the accumulation of dysfunctional CAR-T cells displaying hallmarks of T-cell exhaustion [14,69]. Whereas the combination of CAR-T cells and checkpoint-blocking antibodies (Figure 5) has entered the phase of clinical evaluation in hematological malignancies, the lion’s share of reported data on boosting CAR-T-cell therapy in solid tumors by breaking checkpoint-mediated resistance has been preclinical so far. Seeking to circumvent PD-1-mediated inhibition of CAR-T cells in an orthotopic murine model of pleural mesothelioma, Cherkassky et al. employed a PD-1-blocking antibody to rescue the functionality of mesothelin-specific CAR-T cells [76]. Notably, PD-1 blockade was specifically important for CD28-co-stimulated CAR-T cells owing to greater PD-1 upregulation after antigen encounter as compared with CAR-T cells with 4-1BB costimulation [76]. The necessity of repeated antibody injections emerged as a potential drawback of antibody-dependent PD-1 blockade, prompting genetic engineering projects to create CAR-T cell with an intrinsic PD-1 resistance, for instance, mediated by PD-1 disruption or expression of a dominant negative PD-1 receptor [76] (Figure 5). The former, namely, PD-1 knockout, is already tested in several clinical trials (NCT03525782, NCT03706326, NCT03545815, and NCT03747965), which have not report results yet. Adusumilli et al. reported on first clinical data at the ASCO and AACR meetings in 2019 of mesothelin-specific CAR-T cells combined with a PD-1 blocking antibody in mesothelioma and breast cancer patients. This resulted in very promising 2/14 complete responses, 5/14 partial responses, and 4/14 stable diseases [77]. Similar to PD-1-blocking antibodies, the blockade of PD-L1 could have beneficial effects in the combination with CAR-T cells, as demonstrated by the successful eradication of rhabdomyosarcoma in an orthotopic murine model by novel FGF receptor 4 (FGFR4, CD334)–specific CAR-T cells [78]. However, in this study, CAR-T cells were not solely combined with a PD-L1-blocking antibody but with a whole cocktail of antimyeloid drugs additionally targeting IDO1, iNOS, and TGFβ [78]. Based on the observation of hyperprogression in advanced non–small cell lung carcinoma (NSCLC) treated with PD-1/PD-L1 blocking antibodies, CAR-T cells directly targeting PD-L1 were generated [79]. While NSCLC cells with high PD-L1 expression were consistently eliminated by PD-L1-specific CAR-T cells in xenograft NSCLC mouse models, NSCLC cells with low PD-L1 expression required prior local radiotherapy for efficient clearance [79]. Given the presence of PD-L1 on a variety of healthy tissues [80], direct PD-L1 targeting via CAR-T cells might predispose for on-target/off-tumor toxicity. Therefore, the efficacy and toxicity of PD-L1-specific CAR-T cells should be compared with the combination of CAR-T cells with PD-1/PD-L1 blocking antibodies to elucidate a potential superiority of either approach. 

Advanced pediatric and adult gliomas are tumors with a dismal prognosis given the lack of efficacy displayed by currently available therapies [81]. Hence, the use of CAR-T-cell therapy as a novel therapeutic agent for gliomas, notably glioblastoma, was vigorously investigated. Evidence from preclinical studies in glioma mouse models suggests augmented functionality of tumor-specific T cells in response to PD-1 blockade [82,83]. Moreover, the analysis of resection specimen obtained from glioblastoma patients after treatment with EGFRvIII-specific CAR-T cells revealed, among others, strong PD-L1 upregulation, providing clinical corroboration for the relevance of the PD-1/PD-L1 axis in glioma [84,85]. Hence, early-phase clinical trials enrolling patients with glioma are currently evaluating the combination of CAR-T cells and PD-1 blocking antibodies. In one clinical trial conducted at the University of Pennsylvania, patients with newly diagnosed high-risk glioblastoma received CAR-T cells targeting EGFRvIII combined with pembrolizumab (NCT03726515). In another clinical trial led by investigators from the City of Hope Medical Center, patients with recurrent or refractory glioblastoma will be infused with IL13Rα2-specific CAR-T cells, followed by repetitive nivolumab infusions (NCT04003649). Investigators at the Baylor College of Medicine combined GD2-specific CAR-T cells with a PD-1 blocking antibody for the treatment of neuroblastoma patients (NCT01822652) with limited success; 3/11 patients had a stable disease, and 8/11 had a progressive disease [86]. Notably, PD-1 inhibition combined with lymphodepletion did not further enhance the expansion or persistence of the CAR-T cells compared with lymphodepletion alone [86]. Additionally, in different cancer entities, PD-1 blocking antibodies were used in clinical trials. In claudin 18.2–expressing tumors (e.g., gastric and pancreatic cancer, NCT03874897), the addition of a PD-1 antibody resulted in a favorable interim analysis with 18/37 patients with a partial response and 9/37 patients with a stable disease, paired with a grade 3 or higher hematologic toxicity in all patients and a grade 1 or 2 cytokine release syndrome (CRS) in most patients [87]. Clinical trials with HER2-specific CAR-T cells for the treatment of sarcoma (NCT04995003) and NKG2D-ligand-specific CAR-T cells for the treatment of unresectable metastatic colorectal cancer (NCT04991948) combined with PD-1-blocking antibodies were started approximately 1 year ago and have not reported results yet.

Furthermore, in several clinical trials, CAR-T cells were genetically manipulated to express blocking anti-PD1 or anti-PD-L1 antibodies (NCT02862028, NCT02873390, NCT03182816, NCT03170141, NCT03615313, NCT03030001, NCT03182803, NCT03179007, and NCT03184753), nanobodies (NCT05373147, NCT05089266, NCT04503980, and NCT04489862), or scFvs (NCT04556669, and NCT04842812), which, after secretion by the CAR-T cells, can bind to their ligands (Figure 5). This is an elegant way of PD-1/PD-L1 blocking in a very local fashion, probably preventing unwanted side effects of the systemic application of the blocking antibodies. Unfortunately, no results on these clinical trials have been reported yet. 

In a novel approach to combat CAR-T-cell dysfunctionality, Weber et al. introduced the concept of transiently resting of CAR-T cells to allow for restoring functionality by remodeling epigenetic signatures related to T-cell exhaustion [88]. By the use of dasatinib (Figure 4), a commercially available kinase inhibitor that reversibly shuts down CAR signaling, a transient rest was imposed on CAR-T cells either prior to the onset of T-cell exhaustion or after the acquisition of the exhaustive phenotype [88]. The presence of dasatinib during CAR-T-cell expansion could reduce tonic CAR signaling [88]. As a consequence, a memory-related phenotype of CAR-T cells was preserved, and the functionality of CAR-T cells in vitro and in several xenograft mouse models of solid tumors was improved [88]. To perpetuate CAR-T-cell functionality in vivo by reversing exhaustion-related phenotypes, the transient rest was induced for a couple of days by pulsed application of dasatinib [88]. Remarkably, a single pulse of dasatinib was sufficient to counteract the hallmark features of exhaustion and increase the performance of exhausted CAR-T cells in dasatinib-insensitive mouse models of liquid and solid tumors [88]. Collectively, the use of kinase inhibitors, such as dasatinib, may hold potential to ameliorate exhaustion-induced CAR-T-cell dysfunctionality. However, careful intermittent dosing regimens are required, given the per se CAR-T-cell inhibitory actions of dasatinib [89].

The addition of costimulation to CAR-T cells marked one of the biggest breakthroughs in CAR-T-cell therapy to date. Next, it was demonstrated that superior antitumor activity is achieved by CD28-co-stimulated CAR-T cells coexpressing 4-1BBL (28z-41BBL CAR-T cells) [90]. Combined stimulation through CD28 and 4-1BBL eventuated in increased T-cell persistence and mitigated T-cell exhaustion [90]. Pharmacologically, the design of 28z-41BBL CAR-T cells can be approximated by combining CD28-co-stimulated CAR-T cells with an agonistic 4-1BB antibody (Figure 4). Adjuvant anti-4-1BB therapy improved the cytokine secretion, proliferation, and antitumor activity of HER2-specific CAR-T cells in mouse models of solid tumors [91]. 

#### 2.2.4. Targeting the Tumor Microenvironment

A crucial obstacle to CAR-T-cell therapy in solid tumors is posed by an immunosuppressive tumor microenvironment (TME), the complex architecture and role of which has been extensively reviewed elsewhere [14,69,92]. Transforming growth factor (TGF)-β constitutes a prevalent immunosuppressive agent in the TME capable of neutralizing T-cell activity [89] (Figure 6). Exposure to TGFβ diminishes canonical T-cell functions, such as cytotoxicity, cytokine secretion, and proliferation [93]. With the intention to spare CAR-T cells targeting the receptor tyrosine kinase-like orphan receptor 1 (ROR1) antigen from detrimental TGFβ signaling, the specific kinase inhibitor SD-208 was exploited to block the TGFβ pathway in ROR1-specific CAR-T cells [94]. The pharmacological inhibition of TGFβ could preserve CAR-T-cell functionality and viability in the presence of TGFβ [94]. Overall, the combination of CAR-T cells and pharmacological TGFβ inhibition should be further investigated in in vivo models and clinical studies. Recently, the clinical application of CAR-T cells equipped with a dominant-negative TGFβ receptor (Figure 6) (NCT05166070, NCT05141253, NCT05489991, NCT04227275, NCT03089203, NCT00889954) showcased the feasibility and general safety of conferring TGFβ resistance on CAR-T cells [95]. 

Adenosine, a metabolite generated by adenosine triphosphate (ATP) degradation, constitutes another soluble immunosuppressive factor in the TME [96] (Figure 6). After binding to the adenosine 2a receptor (A2aR), which is upregulated during CAR-T-cell activation and expansion, adenosine was shown to impair the proliferation and killing capacity of CAR-T cells [96,97]. The pharmacological A2aR antagonist SCH-58261 (Figure 6) could level the adenosine-induced deficits in CAR-T-cell proliferation and cytokine secretion but failed to restore CAR-T-cell cytotoxicity [97]. In contrast, genetic targeting via shRNA-mediated gene silencing (Figure 6) decreased the expression of A2aR and imparted complete resistance to adenosine-mediated immunosuppression [97]. In sum, adenosine has emerged as an important inhibitory factor in the TME, and so far, genetic engineering approaches outperformed drug-based approaches in conferring resistance to adenosine-signaling on CAR-T cells.

Through restraining CAR-T-cell infiltration and survival, the TME protects tumor cells from elimination by CAR-T cells. In an orthotopic model of locally advanced breast cancer, the coadministration of CAR-T cells together with the stimulator of the IFN gene (STING) agonist DMXAA led to enhanced tumor control [98] (Figure 7). Mechanistic evidence obtained from single-cell RNA sequencing experiments identified the DMAXX-induced T-cell-attracting chemokines CXCL9 and CXCL10 as critical factors for augmented CAR-T-cell infiltration into the TME [98]. Moreover, DMXAA mediated a depletion of immunosuppressive myeloid cells from the TME, thus further supporting CAR-T-cell infiltration and persistence [94]. Nevertheless, durable complete remission could only be achieved by quadruple treatment with CAR-T cells, DMAXX, a PD-1 blocking antibody, and an antibody directed to GR-1 [98]. This study exemplifies a drug-based multipronged approach to simultaneously address crucial obstacles of CAR-T-cell therapy. In a separate preclinical study, the codelivery of STING agonists could improve the efficacy of CAR-T-cell therapy in syngeneic mouse models of melanoma and pancreatic carcinoma [99]. 1

In addition to small molecules and monoclonal antibodies, conventional chemotherapy has sparked interest as a potential combination partner for CAR-T-cell therapy. While the use of chemotherapeutic agents, such as cyclophosphamide and fludarabine, to deplete endogenous lymphocytes before CAR-T-cell infusion represents a standard procedure, the antiproliferative and apoptosis-inducing effects of chemotherapy pose a challenge to the direct combination with CAR-T-cell therapy. In a xenograft animal model of gastric cancer, monotherapy with either ICAM-1-specific CAR-T cells or paclitaxel, a cytotoxic drug commonly used in gastric cancer, reduced the tumor burden but did not induce long-term survival of tumor-bearing mice [100]. Combination therapy with paclitaxel (Figure 7) administered on days −2, +5, and +12 after CAR-T-cell infusion enabled long-lasting complete remissions and conferred the greatest overall survival advantage [100]. 

Although paclitaxel impedes CAR-T-cell proliferation in vitro, PET/CT imaging conducted 8 weeks after adoptive T-cell transfer confirmed CAR-T-cell presence, indicating successful CAR-T-cell expansion and persistence despite repeated applications of paclitaxel [100]. In addition to additive cytotoxicity, an inhibition of immunosuppressive cells in the TME by paclitaxel treatment was discussed as a possible reason for the therapeutic superiority of combined CAR-T cell and paclitaxel application [100]. In another study focusing on NSCLC, docetaxel (Figure 7), a structural relative to paclitaxel, was shown to promote T-cell infiltration into the tumor complex by inducing the production of the chemokine CXCL11 in the TME through the NF-κB-mediated expression of HMGB1 [101]. Moreover, HER2-specific CAR-T cells exhibited enhanced recruitment into the TME following docetaxel pretreatment due to the augmented expression of HMGB1 and CXCL11, the presence of which is generally associated with improved survival in patients diagnosed with NSCLC [101]. Another strategy to increase CAR-T-cell infiltration into the TME is based on adding the cytotoxic agent oxaliplatin (Figure 7) to the lymphodepletion regimen. By activating macrophages in the TME, oxaliplatin gives rise to the production of T-cell-attracting chemokines, such as CCL5, CXCL16, CXCL9, and CXCL10 [102]. In an autochthonous model of lung adenocarcinoma, pretreatment with oxaliplatin eventuated in the improved recruitment of ROR1-specific CAR-T cells into the TME [102]. Besides, oxaliplatin synergized with anti-PD-L1 therapy to improve the overall performance of ROR1-specific CAR-T cells, leading to prolonged survival and superior tumor control [102]. Antigen spreading can redirect T cells in the TME towards tumor antigens, which bears relevance for achieving complete remissions in CAR-T-cell therapy due to heterogeneous antigen expression in solid tumors [103]. While mesothelin-specific CAR-T cells were capable of eradicating tumor cells with homogenous antigen expression in an immunocompetent mouse model, complete remission could not be attained under conditions of heterogeneous antigen expression with 10% of inoculated tumor cells staining negative for mesothelin [103]. Furthermore, not even the combination of mesothelin-specific CAR-T cells with the pharmacological blockade of PD-1, CTLA-4, TGFβ, and IDO could cure tumors with a proportion of 10% antigen negative cells [103]. Strikingly, low-dose treatment with the alkylating agent cyclophosphamide prior to CAR-T-cell infusion capacitated mesothelin-specific CAR-T cells to eliminate tumors composed of up to 25% antigen-negative cells [103]. Mechanistically, the elimination of a mesothelin-negative tumor cell was traced back to bystander effects emanating from endogenous CD8^+^ T cells [103]. Tumor-associated blood vessels constitute a crucial component of the TME. Aiming at facilitating CAR-T-cell access to the TME by transiently remodeling tumor vasculature, Bocca et al. investigated the combination of GD2-specific CAR-T cells together with the angiogenesis inhibitor bevacizumab (Figure 7) in a xenograft model of neuroblastoma [104]. Monotherapy with either CAR-T cells or bevacizumab did not display significant anti-neuroblastoma activity [104]. However, mice cotreated with GD2-specific CAR-T cells and bevacizumab showed a significant reduction of tumor burden and an extension of survival [104]. Moreover, combination therapy led to a massive infiltration of GD2-specific CAR-T cells into the TME [104]. Taken together, chemotherapeutic agents and the inhibition of angiogenesis render the TME more amenable to CAR-T-cell activity. 

## 3. Conclusions

Despite remarkable success in hematological malignancies, CAR-T-cell therapy is still facing challenging obstacles. Drug-based combination therapies hold potential to counteract major CAR-T-cell challenges posed by antigen shutdown, CAR-T-cell dysfunctionality, and an immunosuppressive TME. Here, we discussed different approaches to combine CAR-T cells with small molecules and antibodies to pharmacologically bolster the power of CAR-T cells. Given a plethora of favorable results from preclinical trials, we firmly advocate the clinical evaluation of drug-based CAR-T-cell combination therapies in both hematological malignancies and solid tumors.

## Figures and Tables

**Figure 1 ijms-24-02342-f001:**
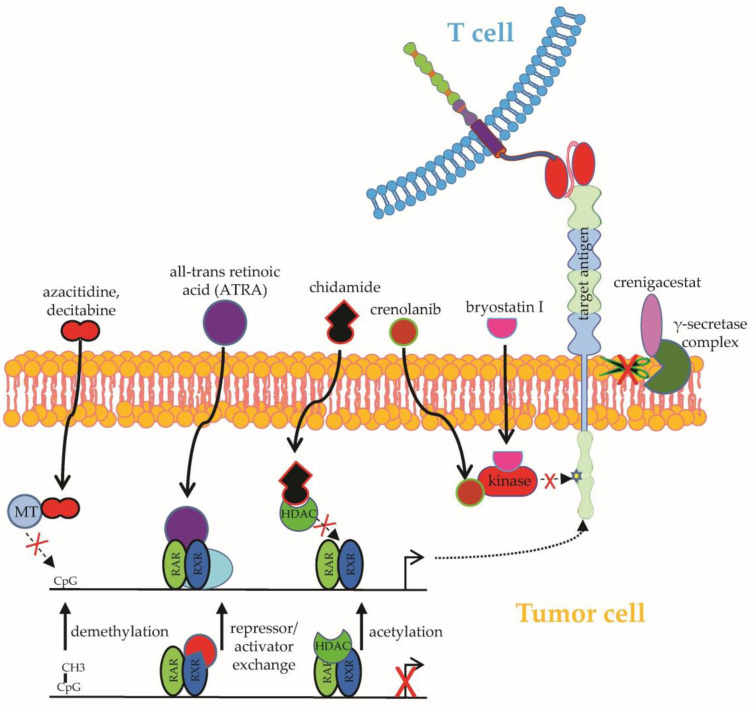
Schematic overview on how to manipulate the target antigen expression on tumor cells. Transcriptional activation of genes encoding target antigens can be induced by: (1) demethylation by methyltransferase inhibitors (e.g., azacitidine or decitabine); (2) exchanging repressor for activator complexes, such as induced by all-trans retinoic acid (ATRA); (3) histone acetylation, such as induced by the HDAC inhibitor chidamide. Prevention of target antigen downregulation on protein level can be induced by: (1) inhibition of cleavage by the γ-secretase complex (e.g., by crenigacestat) and (2) inhibition of target antigen activation and subsequent downregulation by kinase inhibitors (e.g., crenolanib or bryostatin I). MT: methyltransferase; RAR: retinoic acid receptor; RXR: retinoid X receptors. See the main text for further details. The Motifolio Scientific Illustration Toolkit was used for the generation of this figure.

**Figure 2 ijms-24-02342-f002:**
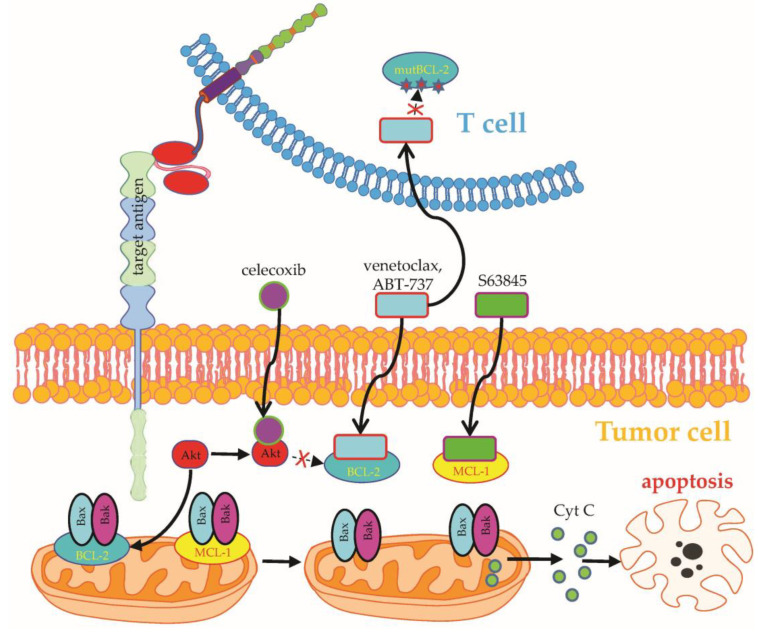
Schematic overview on how to use combinatorial targeting I. CAR-T-cell therapy can be combined with drugs targeting regulators of apoptosis. Examples are the MCL-1 inhibitor S63845 or the BCL-2 inhibitors ABT-737 or venetoclax, which cause the release of cytochrome C from the mitochondria and eventually the apoptosis of the tumor cells. To prevent the effects of these drugs on the CAR-T cells themselves, one can, for example, overexpress mutated BCL-2, which is not able to bind venetoclax, in the CAR-T cells. The COX-2 inhibitor celecoxib also inhibits Akt, causing an inhibition of BCL-2 activation. BCL-2: B-cell lymphoma 2; MCL-1: myeloid cell leukemia 1; Bax: Bcl-2-associated X protein; Bak: Bcl 2 homologous antagonist/killer. See the main text for further details. The Motifolio Scientific Illustration Toolkit was used for the generation of this figure.

**Figure 3 ijms-24-02342-f003:**
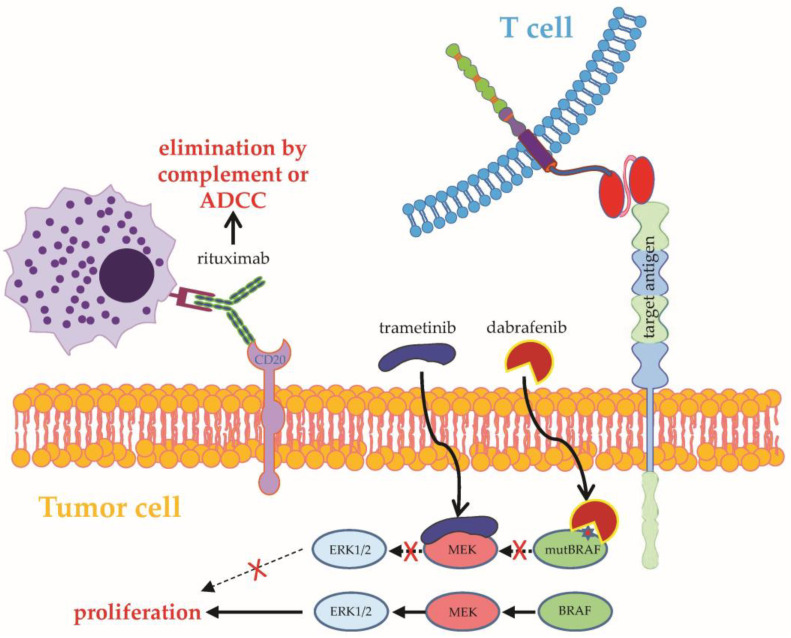
Schematic overview on how to use combinatorial targeting II. CAR-T-cell therapy can be combined with antibodies targeting tumor surface molecules. For example, rituximab targets CD20 and can induce elimination of the tumor cells by complement or ADCC. Furthermore, BRAF and MEK inhibitors (e.g., dabrafenib and trametinib, respectively) can be used to inhibit proliferation of tumor cells. ADCC: antibody-dependent cell-mediated cytotoxicity; BRAF: v-Raf murine sarcoma viral oncogene homolog B; MEK: mitogen-activated protein kinase kinase; ERK: extracellular signal-regulated kinase. See the main text for further details. The Motifolio Scientific Illustration Toolkit was used for the generation of this figure.

**Figure 4 ijms-24-02342-f004:**
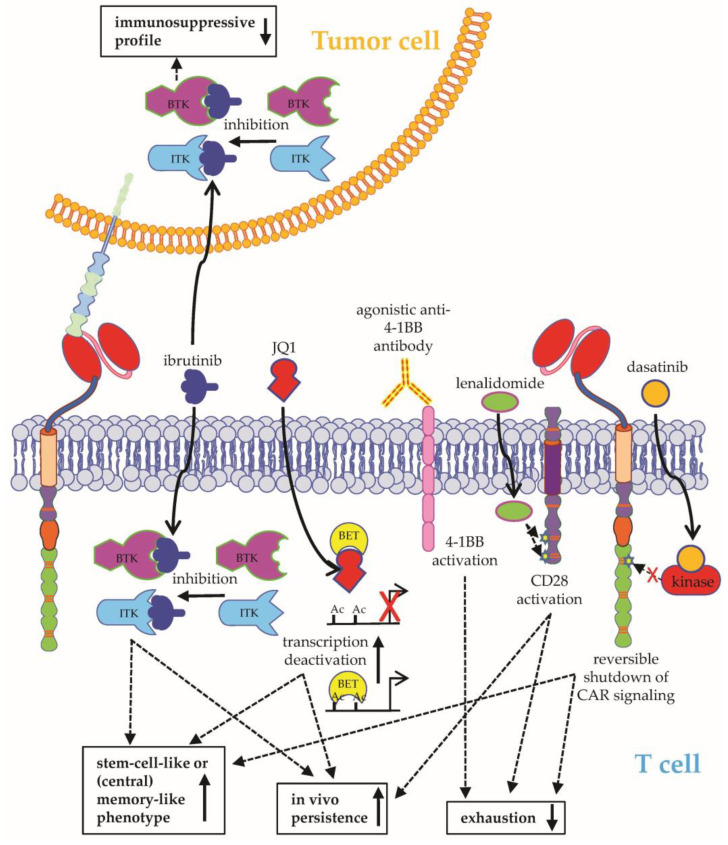
Schematic overview on how to improve CAR-T-cell functionality I. CAR-T-cell functionality can be improved by adding an inhibitor of Bruton’s tyrosine kinase (BTK) and IL-2-inducible T-cell kinase (ITK)(Ibrutinib), an inhibitor of bromodomain and extraterminal motif (BET) proteins (JQ1), an agonistic 4-1BB antibody, an activator of CD28 (lenalidomide), or a kinase inhibitor (dasatinib). See the main text for further details. The Motifolio Scientific Illustration Toolkit was used for the generation of this figure.

**Figure 5 ijms-24-02342-f005:**
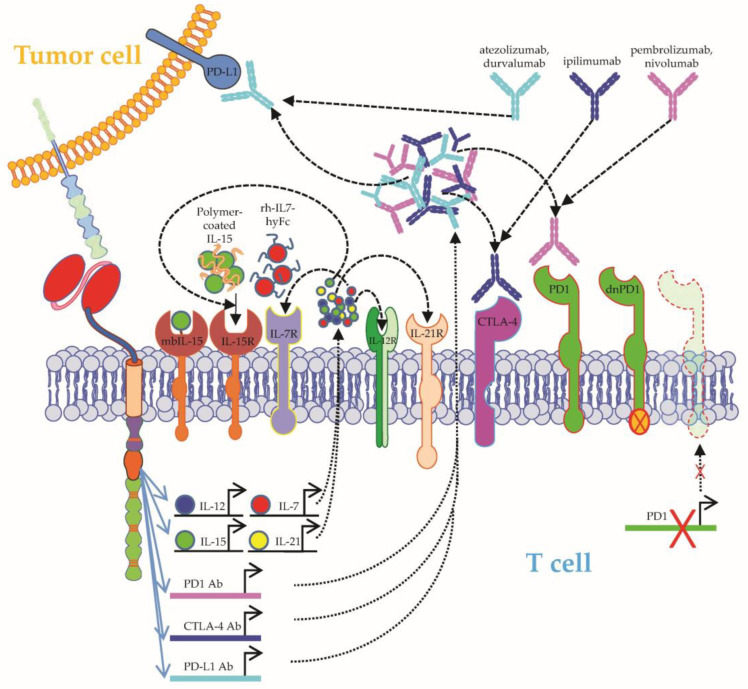
Schematic overview on how to improve CAR-T-cell functionality II. CAR-T-cell functionality can be improved by coapplying cytokines (e.g., long-acting polymer-coated IL-15 or recombinant human IL-7 fused to hybrid Fc (rhIL-7-hyFc)) or by introducing cytokine genes into CAR-T cells, whose proteins are then secreted after CAR activation. Additionally, constructs with membrane-bound IL-15 are described. Furthermore, CAR-T-cell functionality can be improved by combination with checkpoint blockade anti-PD1, anti-PD-L1, or anti-CTLA-4 antibodies, which are coinjected into the patient, or are produced by the CAR-T cells themselves after CAR activation. Moreover, to prevent negative regulation by checkpoints, PD1 on CAR-T cells can by knocked out, or a dominant negative variant of PD1 can be introduced. See the main text for further details. The Motifolio Scientific Illustration Toolkit was used for the generation of this figure.

**Figure 6 ijms-24-02342-f006:**
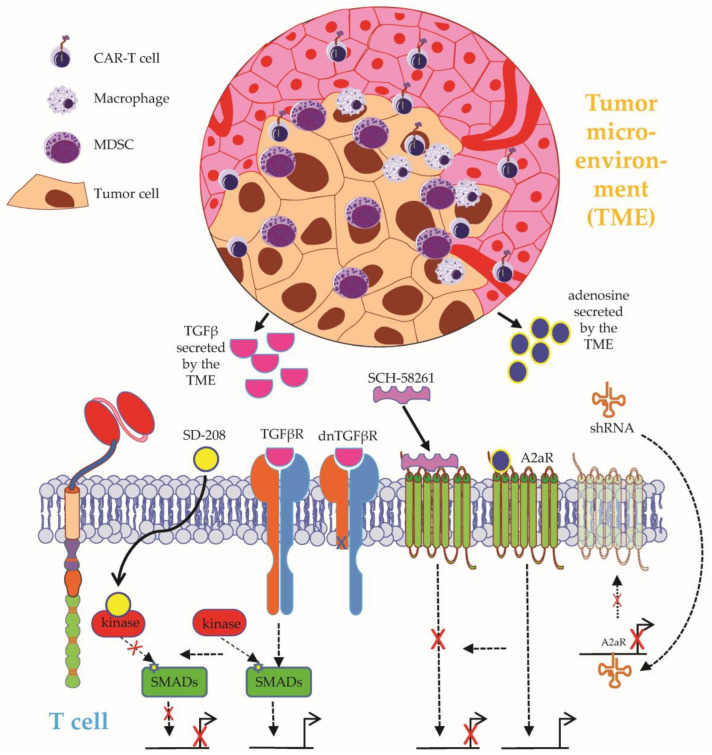
Schematic overview on how to overcome the immunosuppressive tumor microenvironment (TME) I. The negative effects induced by TGFβ secretion by the TME can be overcome by: (1) using the kinase inhibitor SD-2ß8, which will eventually inhibit the TGFβ pathway, or (2) the introduction of a dominant negative form of the TGFβ receptor in the CAR-T cells. The negative effects induced by adenosine secretion by the TME can be overcome by: (1) adding the adenosine 2a receptor (A2aR) antagonist SCH-58261, or (2) adding a short hairpin RNA (shRNA), causing the downregulation of A2aR. MDSC1: myeloid-derived suppressor cell. See the main text for further details. The Motifolio Scientific Illustration Toolkit was used for the generation of this figure.

**Figure 7 ijms-24-02342-f007:**
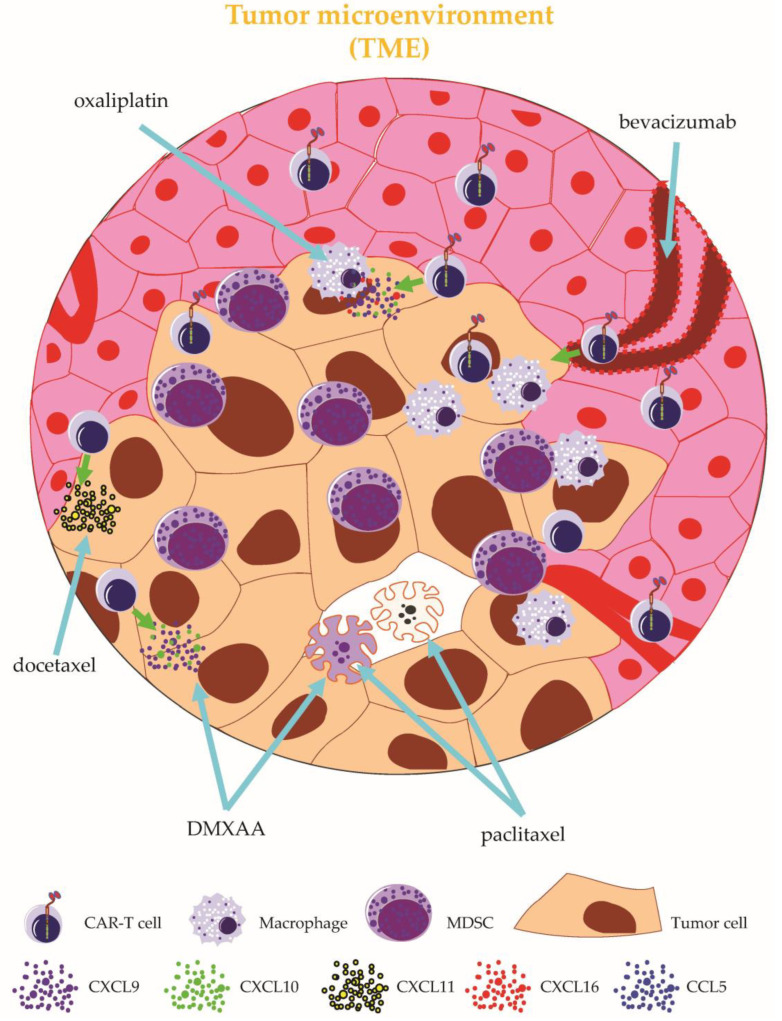
Schematic overview on how to overcome the immunosuppressive tumor microenvironment (TME) II. The immunosuppressive TME can be overcome by: (1) adding the STING agonist DMXAA, causing the death of suppressor cells and the secretion of chemokines, in turn attracting immune cells (e.g., CAR-T cells); (2) adding paclitaxel, causing the death of tumor and suppressor cells; (3) adding docetaxel, causing the secretion of chemokines, in turn attracting immune cells (e.g., CAR-T cells); (4) adding oxaliplatin, causing macrophage activation and the secretion of chemokines, in turn attracting immune cells (e.g., CAR-T cells); and (5) adding the angiogenesis inhibitor bevacizumab, causing an improved extravasation of CAR-T cells from the blood into the tumor. MDSC: myeloid-derived suppressor cell. See the main text for further details. The Motifolio Scientific Illustration Toolkit was used for the generation of this figure.

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
