# Peer review of "CARs and Drugs: Pharmacological Ways of Boosting CAR-T-Cell Therapy"

_ijms, 2023, doi:10.3390/ijms24032342_

Round 1

Reviewer 1 Report

The authors have done a great job putting together a review paper on improving the efficacy of CAR-T by pharmacological means. 

- There are multiple agents implicated in various sections, which act by increasing the antigen expression, however can we add a review of current agents which are in clinical trials or the reasons why didn't they make to clinical trials? A table with list of phase II/III clinical trial will add a lot to the article and make it more interesting for the audience.

Reviewer 2 Report

Undoubtedly a topical article. I found the topic very interesting and I also liked the way it was structured. I found the article comprehensive and well written.

Reviewer 3 Report

The review manuscript titled as “CARs and Drugs: Pharmacological ways of boosting CAR-T-cell therapy” submitted by Dr. Harrer, et al, described that CAR-T-cell therapy was facing some obstacles, such as antigen-shutdown, CAR-T-cell dys-functionality and immunosuppressive tumor microenvironment, and addressed that drug-based combination therapies held potential to counteract these CAR-T-cell challenges. These authors also discussed many kinds of approaches to utilize small molecules or antibodies to bolster the power of CAR-T cells. Intriguingly, the manuscript presented some concise schematic images to generalize the effects of pharmacological molecules or antibodies on CAR-T therapy. The manuscript is significant in the field of immunotherapy on how to conquer the obstacles of CAR-T therapy, and interesting to readers who are focusing on cancer treatment.

Reviewer 4 Report

This is a review of CAR-T therapy and how it is boosting, with a focus on pharmacological details.

The figures are all very clear, and the efforts made are well received. In addition, the hematological malignancies are becoming quite generalized and easy to understand, and the presentation of the overall picture of CAR-T or immunotherapy in solid tumors is commendable.

There are no discrepancies in the description, and this review is well regarded as it stands.